# Exploring the Potential of Flow-Induced Vibration Energy Harvesting Using a Corrugated Hyperstructure Bluff Body

**DOI:** 10.3390/mi14061125

**Published:** 2023-05-26

**Authors:** Yikai Yuan, Hai Wang, Chunlai Yang, Hang Sun, Ye Tang, Zihao Zhang

**Affiliations:** 1School of Mechanical Engineering, Anhui Polytechnic University, Wuhu 241000, China; yyikai666@gmail.com (Y.Y.); hang961203@163.com (H.S.); tangye2010_hit@163.com (Y.T.); zihao_0216@163.com (Z.Z.); 2Anhui Key Laboratory of Advanced Numerical Control & Servo Technology, Wuhu 241000, China; 3Anhui Polytechnic University Industrial Innovation Technology Research Co., Ltd., Wuhu 241000, China

**Keywords:** flow-induced vibration, energy harvesting, hyperstructure, bluff body

## Abstract

Fluid-induced vibration is a common phenomenon in fluid–structure interaction. A flow-induced vibrational energy harvester based on a corrugated hyperstructure bluff body which can improve energy collection efficiency under low wind speeds is proposed in this paper. CFD simulation of the proposed energy harvester was carried out with COMSOL Multiphysics. The flow field around the harvester and the output voltage in different flow velocities is discussed and validated with experiments. Simulation results show that the proposed harvester has an improved harvesting efficiency and higher output voltage. Experimental results show that the output voltage amplitude of the harvester increased by 189% under 2 m/s wind speed.

## 1. Introduction

With the widespread usage of miniature, low-power devices such as microelectromechanical systems, wireless sensor networks, and the Internet of Things (IoT) [1,2,3], traditional methods of energy supply are insufficient to meet the demands. As a result, the development of a sustainable power supply for low-energy consumption products has become crucial. Piezoelectric energy harvesters, which use piezoelectric materials and have a straightforward structure that is easily adjustable, less affected by environmental factors, and fully satisfies the needs of the miniaturized market, maximize the conversion of flow-induced vibration into electrical energy. However, the broad frequency ranges from environmental vibrations present a challenge, as traditional piezoelectric energy harvesting systems have comparatively limited operational bandwidth, thus restricting the harvesting efficiency of vibration energy. Therefore, structural optimization of the energy harvester device is often necessary to achieve high-efficiency energy conversion. In recent years, flow-induced vibration piezoelectric energy harvesting (FIVPEH) [4,5,6,7] has emerged as an environmentally friendly, sustainable, and efficient energy harvesting technology that is gaining popularity among researchers. In light of the successful application of hyperstructures in various fields, integrating hyperstructures into flow-induced vibrations may be a possible method to improve the performance of harvesters.

There are various types of energy that exist in natural environments, including vibration, solar, and wind energy. Wind energy can be harvested through flow-induced vibration. Currently, there are three primary methods for harvesting energy from flow-induced vibrations: piezoelectric, electrostatic, and electromagnetic. Among these, piezoelectric energy harvesting devices are more effective due to their small size, simple configuration, low environmental requirements, lack of thermal effects, and high energy density. Piezoelectric devices are also immune to electromagnetic waves, making them highly adaptable. In order to improve the efficiency of flow-induced vibration energy harvesting, various schemes have been proposed by researchers. For example, Sun [8] suggested using a bulb-shaped bluff body to enhance energy harvesting by combining the advantages of vortex-induced and galloping vibrations. Wang [9] conducted theoretical and practical studies on the efficiency of energy harvesting devices using circular and square sections of a bluff body at three different attack angles. In addition, Wang [10] made a discovery about how to combine metasurface structures with flow-induced vibrations, designed four periodic metasurfaces, and examined their influence on energy harvesting efficiency. Zhang [11] investigated the impact of side ratio and load resistance on the onset velocity, displacement, and power output of a linear-based energy harvester. Zhang [12] explored the influence of the Reynolds number on piezoelectric energy harvesting from the vortex-induced vibrations (VIVs) of a circular cylinder. Additionally, Zhang [13] examined the effects of different attack angles on the energy harvester.

A flow-induced vibration energy harvester based on a corrugated hyperstructure bluff body, which consists of a corrugated hyperstructure bluff body and a piezoelectric cantilever beam attached, is proposed in this paper. The harvesting efficiency can be improved by introducing a hyperstructure into the bluff body, which is validated through numerical simulation and experiments. The performance of the proposed harvester under different wind speeds is discussed.

## 2. Flow-Induced Vibration Harvester Based on a Corrugated Hyperstructure Bluff Body

### 2.1. Structure Design

The schematic diagram of a conventional galloping energy harvester (GPEH) is shown below in Figure 1. The flow-induced vibration is harvested through the piezoelectric sheet attached to the fixed end of the cantilever beam. As is shown in Figure 1, one end of the cantilever beam is fixed, while the other end is attached to the bluff body. When the wind flows through the bluff body, an excitation force is generated due to the aerodynamic properties of the flow field around the bluff body. The excitation force causes the bluff body to oscillate, which in turn causes the cantilever beam to oscillate. The oscillation of the cantilever beam results in the conversion of mechanical energy into electrical energy through the piezoelectric effect in the piezoelectric sheet.

Smooth bluff bodies are used in a conventional GPEH. A novel galloping energy harvester based on the corrugated bluff body is proposed in this paper to improve the harvesting efficiency. As is shown in Figure 2, regularly spaced raised structures are integrated on the surface of the bluff body to form a corrugated bluff body. The hyperstructure on the bluff body surface can tune the aerodynamic properties of the bluff body, which can make the bluff body vibrate easily and stably. The corrugated bluff body offers practical benefits beyond aerodynamic efficiency. Its modular design enables easy installation and customization in various environments while also ensuring durability and resistance to environmental damage.

As is shown in Figure 2, a cantilever with a piezoelectric patch is connected to the bluff body. *L* and *W* are the height and width of the bluff body, and *L_b_*, *W_b_*, and *h_b_
*are the cantilever beam length, width, and thickness.

### 2.2. Mathematical Model of the Cantilever in the GPEH with a Corrugated Bluff Body

When the GPEH with a corrugated bluff body is installed into a flow channel, the bluff body is vibrated perpendicular to the airflow direction by the aerodynamic force, which results in vibration of the attached cantilever. The Euler–Bernoulli beam equation for the vibration cantilever is shown below:(1)$$\frac{\partial^{2}}{\partial x^{2}}\left(EI\frac{\partial^{2}u}{\partial x^{2}}\right)=q$$
where *u* is the displacement, $\frac{\partial u}{\partial x}$ is the slope of the beam, $EI\frac{\partial^{2}u}{\partial x^{2}}$ is the bending moment of the beam, and *q* is the distributed load on the beam.

The mathematical model is established based on the Euler–Bernoulli equation, and the dynamic equation of the GPEH with a corrugated bluff body can be described as follows:(2)$$m\ddot{w}+c\dot{w}+EIw^{{\prime \prime \prime }^{\prime }}{}^{{\left\{w^{\prime }{\left(w^{\prime }w^{\prime \prime }\right)}^{\prime }\right\}}^{\prime }{\left\{\theta \left(1+\frac{1}{2}w^{\prime 2}\right)\right\}}^{\prime \prime }}v\;=F_{galloping}\left(t\right)\left(\delta \left(x-L_{b}\right)-\frac{W}{2}\delta^{\prime }\left(x-L_{b}\right)\right)$$
where *w* is the lateral displacement of the cantilever beam, *m* is the mass of the cantilever beam, *EI* is its bending stiffness, *L_b_* is its length, *c* is the damping coefficient of the system, *θ* is the electromechanical coupling coefficient, and *W* is the width of a bluff body. Additionally, the aerodynamic force generated by the bluff body can be expressed as follows:(3)$$F_{galloping}\left(t\right)=0.5\rho S_{c}U^{2}C_{FZ}$$
where *ρ* is the air density, *U* is the wind speed, and *S_c_ = L × W*, where *L* and *W* are the length and width of the bluff body, respectively. *C_FZ_* is the coefficient of the vertical component of the fluid-dynamic force, and its expression is shown below:(4)$$C_{FZ}=-\left(C_{L}+C_{D}\tan\partial \right)\sec\partial \approx S_{1}\alpha -S_{3}\alpha^{3}$$
where *C_L_* is the lift coefficient, *C_D_* is the drag coefficient, *α* is the attack angle, and *S*_1_ and *S*_3_ are factors in the Taylor approximation.

## 3. CFD Modeling of the Corrugated Bluff Body

### 3.1. Structure of the Corrugated Bluff Body

The structure of the corrugated bluff body with periodic protrusions is shown in Figure 3. The distance between the convex structures of the corrugated bluff body is 6 mm. The shape of the protruding structure is three-quarters of a circle, and the depth of the semicircular groove is 3 mm. A cuboid with a 14 mm × 14 mm cross-section is removed from the center of the corrugated bluff body.

### 3.2. Two-Dimensional CFD Simulations of the Corrugated Bluff Body

Due to the vibration being parallel to the cross section of the bluff body with a constant area, a two-dimensional numerical model was used. The parameters of the corrugated bluff body and the initial conditions of simulation are listed below in Table 1.

As is shown in Table 1, *U* is the velocity of the inlet flow; *W*_1_ is the width of the river basin; *L*_1_ is the length of the river basin; *W*_2_ is the width of the bluff body; and *L*_2_ is the length of the bluff body. Air is used as the fluid in the simulation. *μ* is the aerodynamic viscosity coefficient of the air, and *ρ* is the density of the air. *V_inlet_* is the inlet velocity and *P_outlet_* is the outlet pressure.

As is shown in Figure 4, a rectangular computational area was used in this study, and the flow field size was 26.875*L*_1_ × 5*L*_1_. The distance between the corrugated bluff body and inlet was 3.125*L*_1_. The distance from the corrugated bluff body to the upper and lower walls of the fluid channel was 2.5*L*_1_. The distance was set to 23.75*L*_1_ between the outlet and the bluff body to realize the full development of the vortex streets.

The inlet velocity was set at the inlet boundary, the outlet pressure was set at the outlet boundary, and the wall condition was set as the anti-slip wall.

### 3.3. Validation and Analysis of Simulations

The Navier–Stokes equation is well suited for fluid–structure interaction (FSI) systems, which is defined as follows:(5)$$\rho \frac{DV}{Dt}=\rho f-⛛p+\mu {⛛}^{2}V$$
where *DV/Dt* is the material derivative, *ρ* is the density of the fluid, *V* is the velocity of the fluid, *μ* is the aerodynamic viscosity coefficient of the fluid, *p* is the hydrostatic pressure, and *f* is the external force per unit volume.

To analyze the external force of GPEH with corrugated bluff body, the pressure distribution around the corrugated bluff body needs to be calculated. The lift and drag forces can be expressed as follows:(6)$$F_{lift}=\mathrm{intop}1\left(spf.T\_stressy\right)$$
(7)$$F_{drag}=\mathrm{intop}1\left(spf.T\_stressx\right)$$
where intop1 is an introduced integral operator, *spf*.*T_stressy* is the total stress on the surface of the bluff body along the *y*-axis direction, and *spf*.*T_stressx* is the total stress on the surface of the bluff body along the *x*-axis direction.

A traditional smooth bluff body was used in simulation for comparison, which is shown in Figure 5a. The numerical model of the corrugated bluff body is shown in Figure 5b. The inlet flow velocity was set as U = 1 m/s in both cases. The initial conditions were the same in both cases, and are listed in Table 2.

An implicit backward difference method was applied in the CFD simulation. Extremely refined meshes, as shown in Figure 6, were used for both cases to improve the accuracy of the calculation. The number of mesh elements was 29,834 and 30,921 for each case.

As is shown in Figure 7, the maximum of the lift force generated in the corrugated bluff body was 0.7577 N/m, which increased by 18.3% compared to the traditional bluff body. The average lift force generated in the corrugated bluff body was 0.27032 N/m, which increased by 6% compared to the traditional bluff body.

As is shown in Figure 8, the maximum drag force generated in the corrugated bluff body was 0.87 N/m, which decreased by 18.3% compared to the traditional bluff body. The average drag force generated in the corrugated bluff body was 0.495348 N/m, which decreased by 16.78% compared to the traditional bluff body. The first peak of the lift force of the corrugated bluff body was 0.59 N/m at 2.4 s, while it was 0.435 N/m at 3.82 s in the traditional bluff body, which means vibration occurred earlier in the corrugated bluff body with a large lift force, resulting in large deformations of the cantilever attached to the corrugated bluff body.

As is shown in Figure 9, Föppl vortices were generated around the bluff bodies, which is called the Karman vortex street phenomenon.

The flow velocity of the point with a position at (2200, 400) in the downstream of the vortex is shown in Figure 10.

The maximum flow velocity was 2.75 m/s with the corrugated bluff body, which increased by 41.02% compared to the smooth bluff body.

As is shown in Figure 11, the maximum amplitude of the wake flow with the corrugated bluff body was 0.467 m, indicating a 72% increase compared to the smooth bluff body.

## 4. Prototypes and Experimental Setup

As is shown in Figure 12, a prototype of the galloping energy harvester with a corrugated bluff body was used for validation. The length, width, and thickness of the cantilever beam in the galloping energy harvester were 180 mm, 20 mm, and 1.5 mm, respectively. The cantilever beam and bluff body were made of brass material and ABS, respectively. The bluff body was fabricated by 3D printing. A PZT-5H patch was attached at the fixed end of the cantilever beam.

The parameters of PZT-5H are listed in Table 3.

The schematic diagram of the experimental platform is shown in Figure 13, which consists of a wind tunnel with a speed controller, a GPEH with a corrugated bluff body, and a data acquisition system. The GPEH is mounted on a fixture and placed outside the wind tunnel. The length of the corrugated bluff body is *L* = 120 mm, and the width is *W* = 30 mm. The mass of the corrugated bluff body is 280 g. The size of the PZT-5H film is *L_b_ × W_b_ × h_b_* = 20 mm × 20 mm × 0.2 mm.

The wind tunnel experimental platform is shown in Figure 14. The cross-section of the wind tunnel chamber is 40 cm × 40 cm. The wind flow is supplied by an axial fan installed at the end of the wind tunnel, which is regulated by a unidirectional fan governor (YY-FTQS, Produced by Delixi Company, Shanghai, China). A digital oscilloscope (DS1102E, PUYUAN TECHNOLOGY CO., LTD., Beijing, China) was used to measure and record the output voltage across the load resistance connected to the PZT film.

The tuning range of the wind speed in the wind tunnel was 0 m/s ≤ U ≤ 6 m/s. Based on the formula *Re = ρvd/μ*, the Reynolds number for the wind tunnel experiment was calculated to be approximately 2162. The output voltages of the PZT film attached to the cantilever of the GPEH with a corrugated bluff body were measured by the data acquisition system under different wind speeds. The galloping energy harvester with a smooth square cross-section bluff body was tested for comparison.

## 5. Experimental Results and Discussions

The PZT film was set to an open circuit during experiments. The output voltage of the PZT film under different wind speeds is shown in Figure 15.

As is shown in Figure 15, the output voltage of the PZT film increased with the wind speed in both GPEHs. The vibration of the cantilever of the GPEH with a corrugated bluff body began at about 1 m/s wind speed with a maximum output voltage of 0.03905 V. When the wind speed increases to 6 m/s, the maximum output voltage can reach 0.8905 V.

At a wind speed of 2 m/s, the average output voltage and power of the PZT film in the GPEH with a corrugated bluff body were 0.11 V and 0.03133 W, respectively, which increased by 182% and 36.2% compared to the GPEH with a smooth bluff body. The GPEH with a corrugated bluff body can realize higher harvesting efficiency at low wind speeds, which improves the performance of the GPEH in low wind conditions.

As is shown in Figure 16, a “lock-in” area is produced during wind speeds from 2 m/s to 3 m/s, and the output voltage of the PZT film tends to decrease and then increase. This means that the harvesting efficiency of the GPEH based on the corrugated bluff body is improved under low wind speeds. When the critical value of wind speed *U* = 3 m/s is exceeded, the output voltage will continue to rise.

The peak output voltages of the GPEH with a corrugated bluff body and smooth bluff body at different wind speeds are shown in Figure 17. The peak voltage of the GPEH with a corrugated bluff body was 0.45898 V at 2 m/s wind speed, which increased by 189% compared to the GPEH with a smooth bluff body.

The frequency spectrum of the output voltages of the corrugated bluff body and the smooth bluff body at a wind speed of *U* = 2 m/s is shown in Figure 18.

As is shown in Figure 18, the first resonant frequency of the galloping energy harvester with a corrugated bluff body was 3.73 Hz with a 0.7391 Hz bandwidth. The average voltage in the bandwidth was about 0.1158 V. However, the galloping energy harvester with a smooth bluff body had two peaks in the 2–6 Hz frequency range, which were 3.24 Hz and 3.86 Hz. The bandwidths of the above two peaks were 0.33184 Hz and 0.39131 Hz with average output voltages of 0.0052 V and 0.0054 V. With the introduction of the metastructure to the bluff body, the average output voltage of the galloping energy harvester increased 20 times compared to that of the traditional one. Meanwhile, the bandwidth of the galloping energy harvester with a corrugated bluff body increased by 2.2% compared to that of the traditional one in the 2–6 Hz range.

## 6. Conclusions

A GPEH with a corrugated bluff body meant to improve the harvesting efficiency of fluid-induced vibration was proposed in this paper. CFD simulation results of the corrugated bluff body show that the maximum drag force can reach 0.87 N/m, which is an increase of 18.3% compared to that of the traditional one. The peak of the lift force generated in the corrugated bluff body was 35.63% higher than that of the traditional one.

The performance of the GPEH with the corrugated bluff body was validated on an experimental platform. Experimental results show that the maximum output voltage of the GPEH with a corrugated bluff body can reach 0.45893 V with an average power of 0.03133 W, which is 189% and 36.2% higher than that of the traditional GPEH, respectively. The “lock-in” range of the output voltage of the GPEH with a corrugated bluff body occurs during wind speeds from 2 m/s to 3 m/s, which improves the harvesting efficiency at low wind speeds. Simulation and experimental results demonstrate that the GPEH with the corrugated bluff body has a high harvesting efficiency in low wind speed conditions compared to a traditional one, providing a possible solution for future applications of wind energy harvesters based on flow-induced vibrations.

## Figures and Tables

**Figure 1 micromachines-14-01125-f001:**
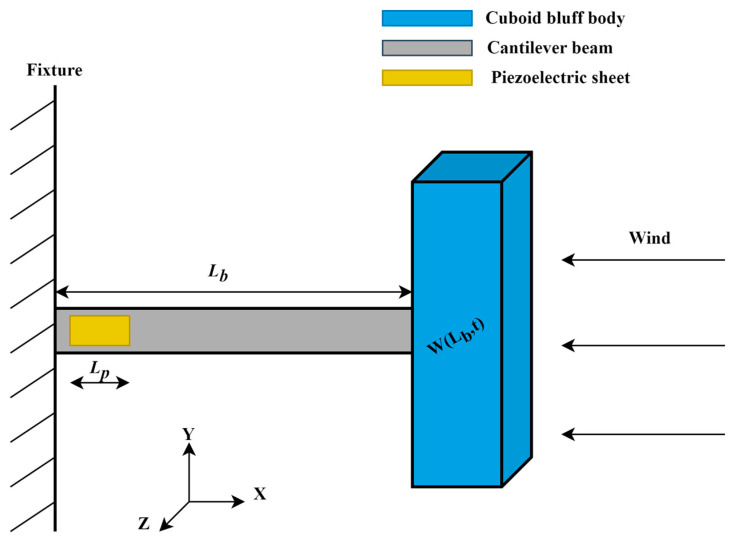
Schematic diagram of a typical GPEH.

**Figure 2 micromachines-14-01125-f002:**
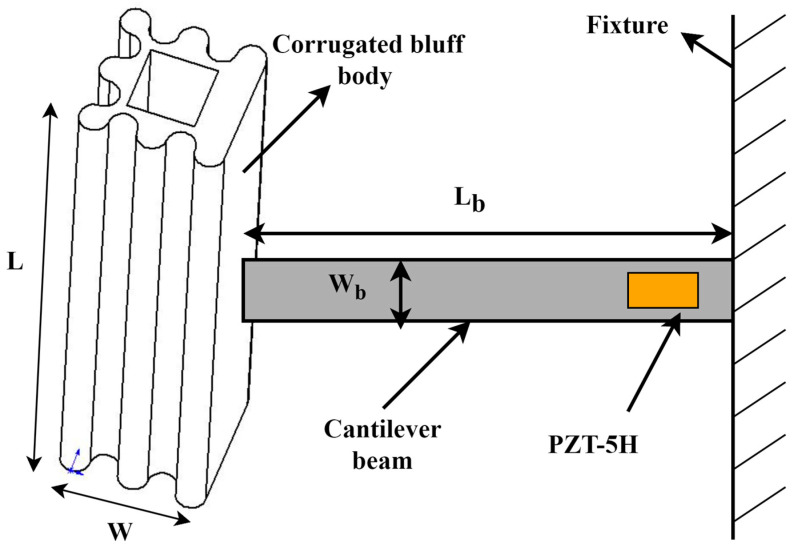
The 3D schematics of the wind energy harvester with a corrugated bluff body.

**Figure 3 micromachines-14-01125-f003:**
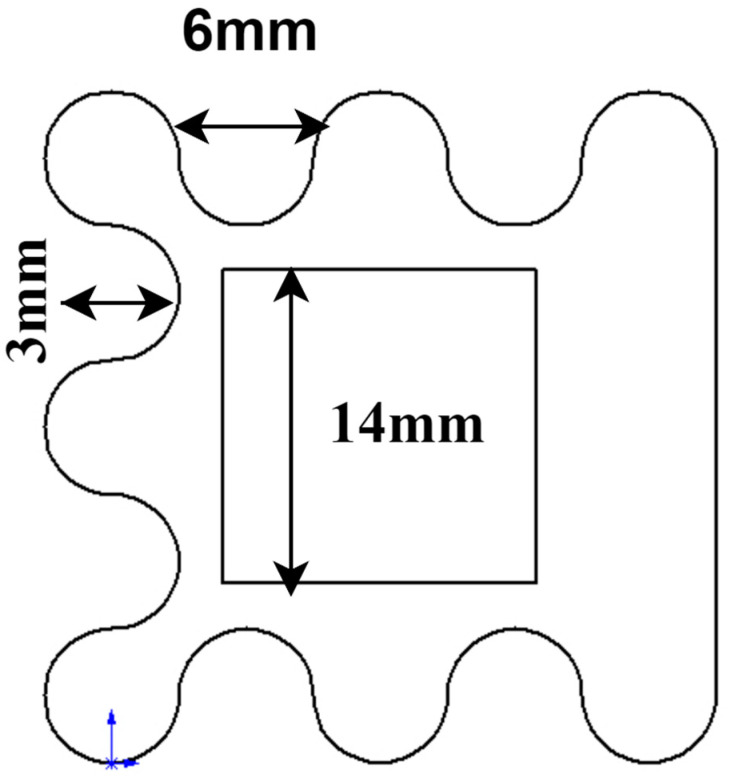
Top graph of corrugated bluff body.

**Figure 4 micromachines-14-01125-f004:**
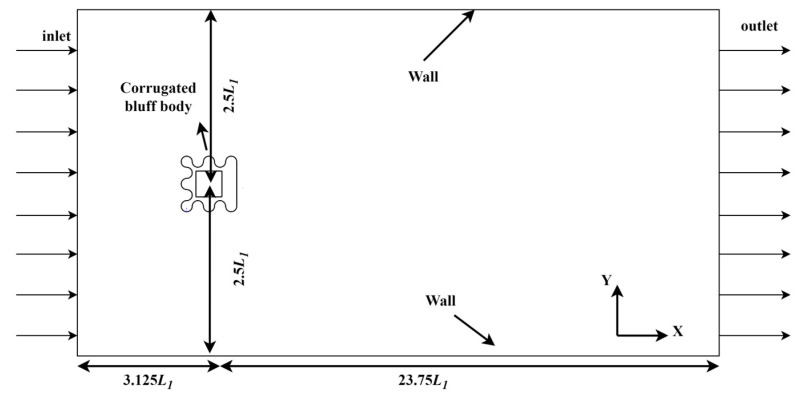
System computational domain settings.

**Figure 5 micromachines-14-01125-f005:**
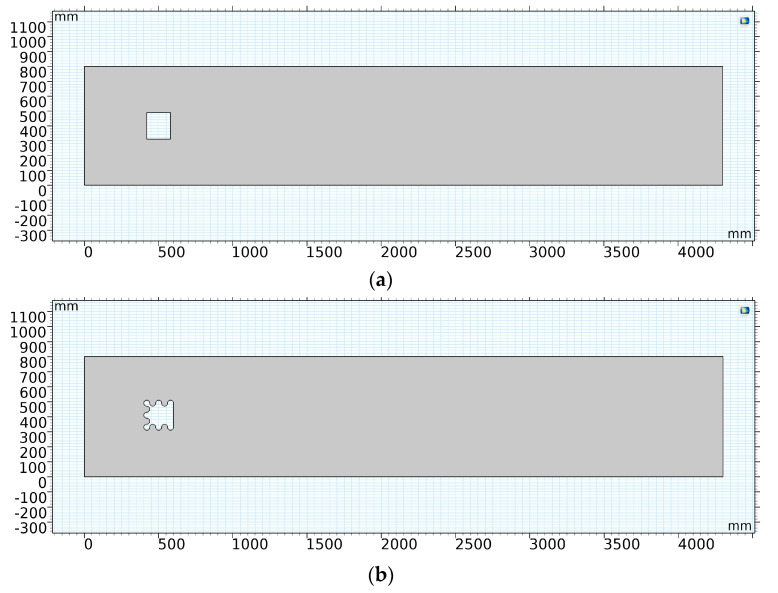
Numerical models of traditional and corrugated bluff body. (**a**) Numerical model of the traditional bluff body. (**b**) The numerical model of the corrugated bluff body.

**Figure 6 micromachines-14-01125-f006:**
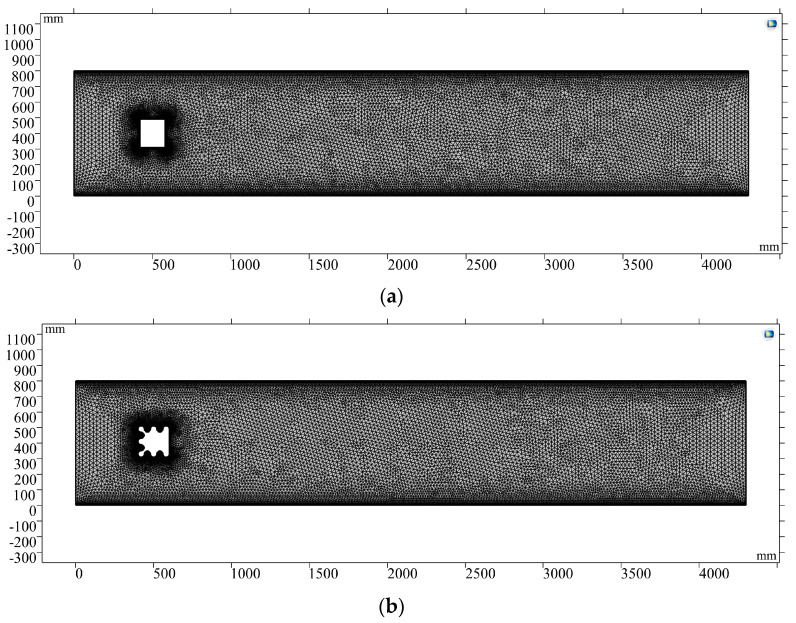
Mesh of CFD models.(**a**) Mesh of CFD model with the traditional bluff body. (**b**) Mesh of CFD model with the corrugated bluff body.

**Figure 7 micromachines-14-01125-f007:**
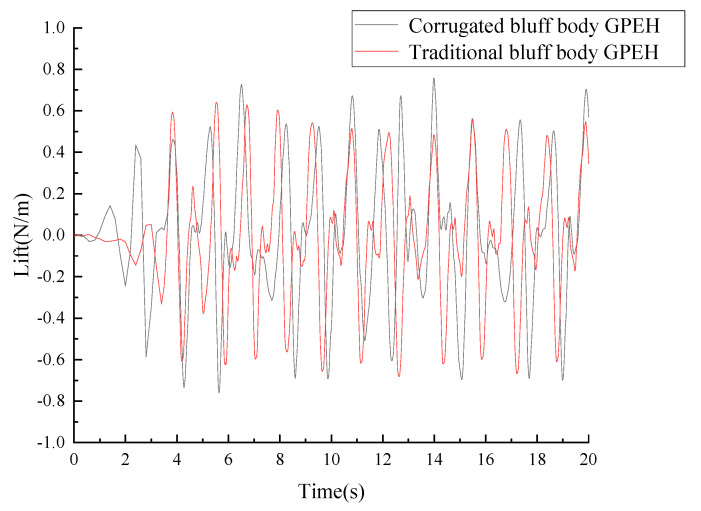
Lift forces of different GPEHs (*U* = 1 m/s).

**Figure 8 micromachines-14-01125-f008:**
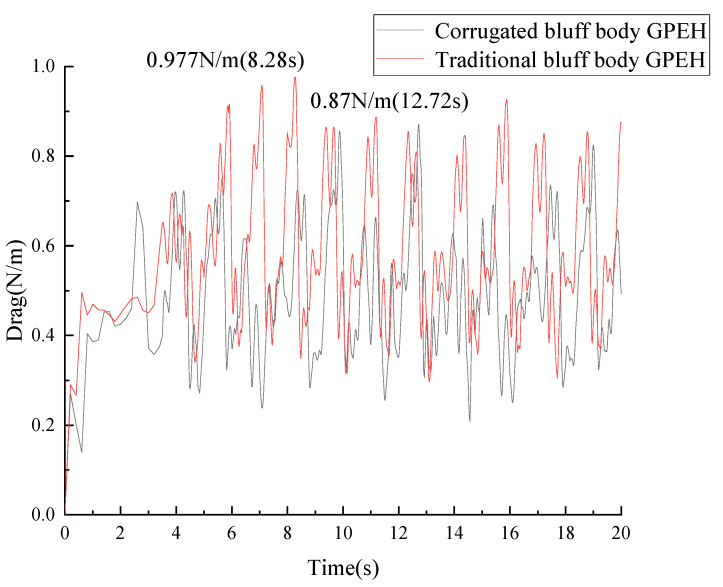
Drag forces of different GPEHs (*U* = 1 m/s).

**Figure 9 micromachines-14-01125-f009:**
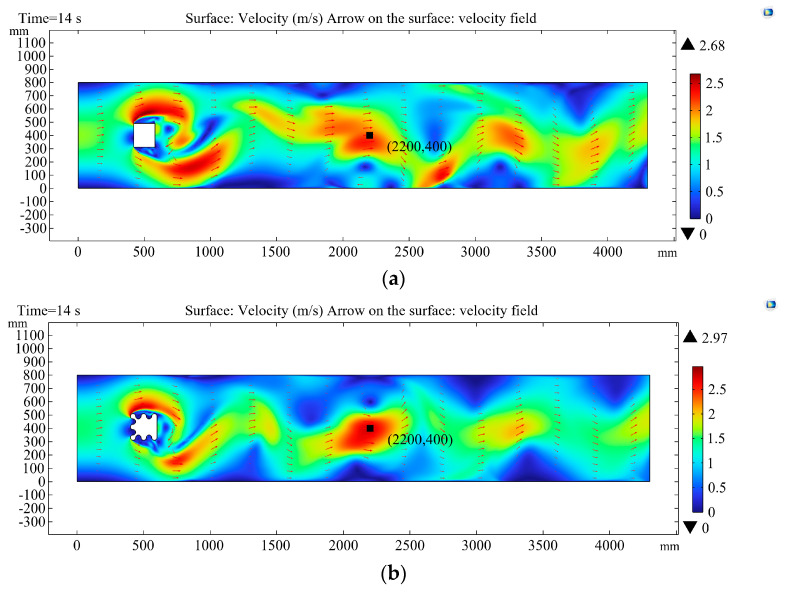
Velocity cloud maps of traditional and corrugated bluff body. (**a**) Velocity cloud maps of the traditional bluff body. (**b**) Velocity cloud maps of the corrugated bluff body.

**Figure 10 micromachines-14-01125-f010:**
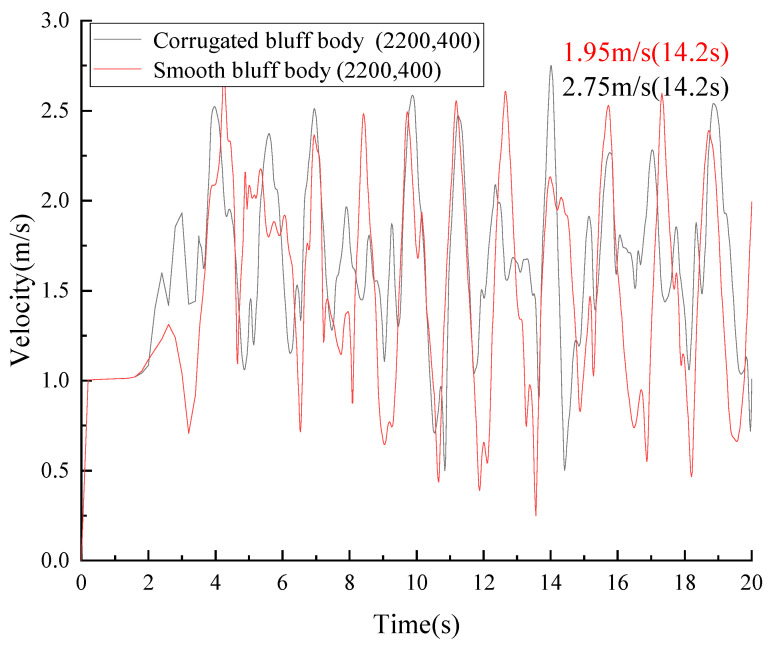
Velocity of the wake basin (2200, 400).

**Figure 11 micromachines-14-01125-f011:**
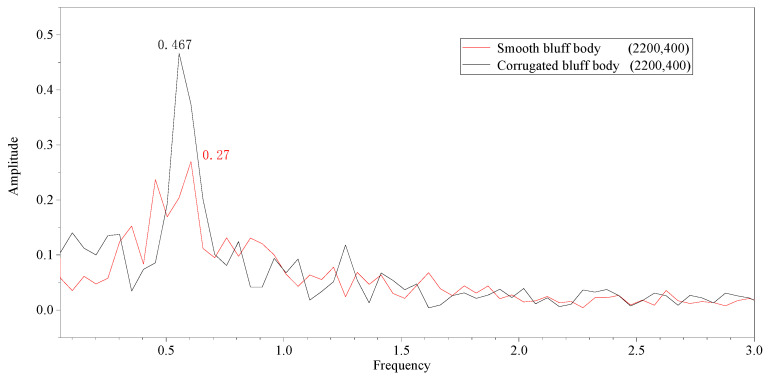
The frequency diagram of the wake basin (2200, 400).

**Figure 12 micromachines-14-01125-f012:**
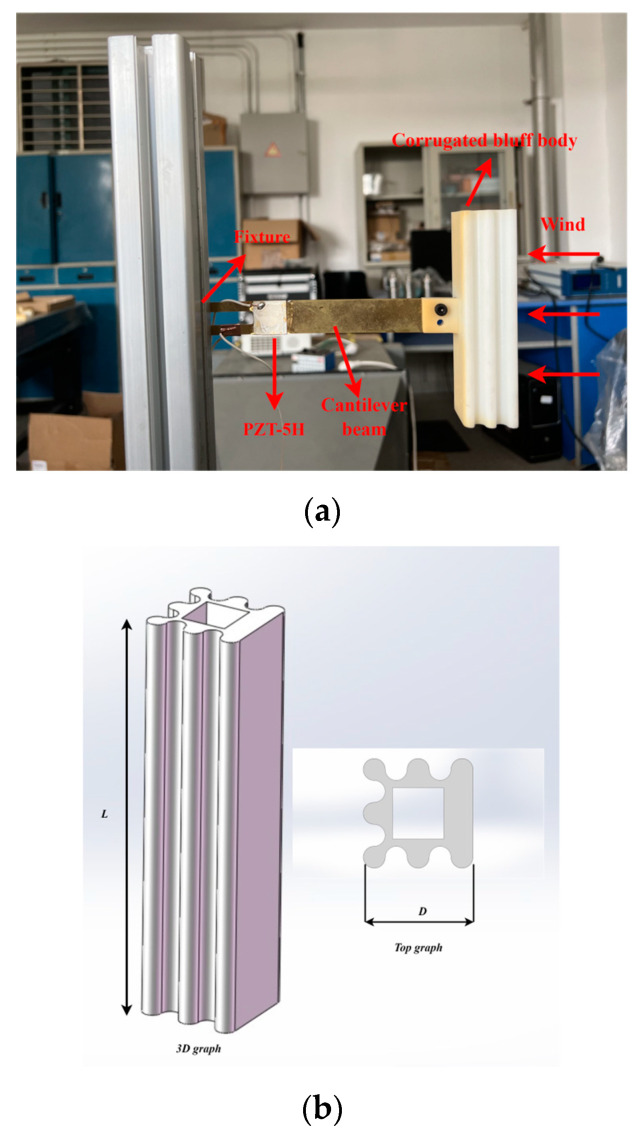
Prototype of corrugated bluff body. (**a**) Picture of real products. (**b**) Three-dimensional diagram of corrugated bluff body.

**Figure 13 micromachines-14-01125-f013:**
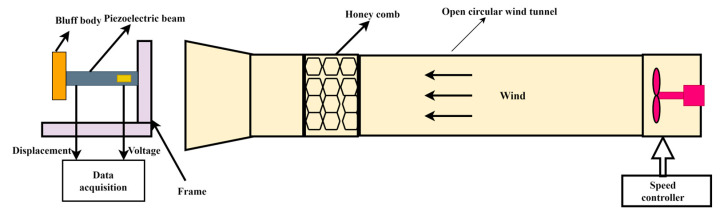
Schematic diagram of wind tunnel experimental platform.

**Figure 14 micromachines-14-01125-f014:**
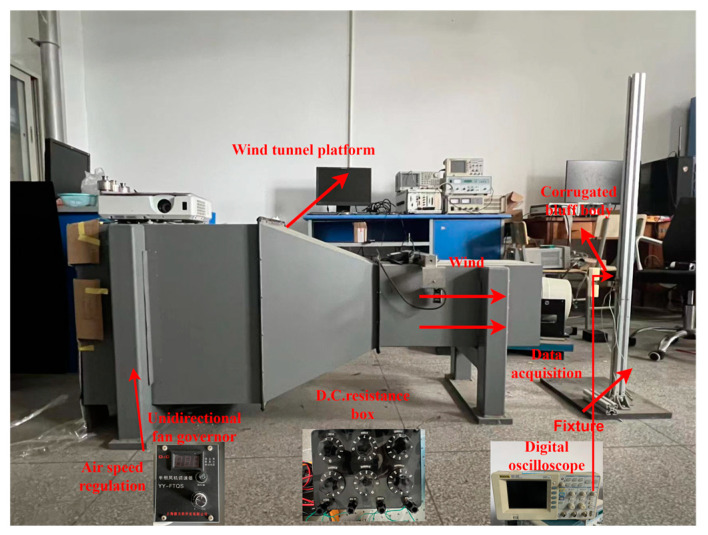
The wind tunnel experimental platform.

**Figure 15 micromachines-14-01125-f015:**
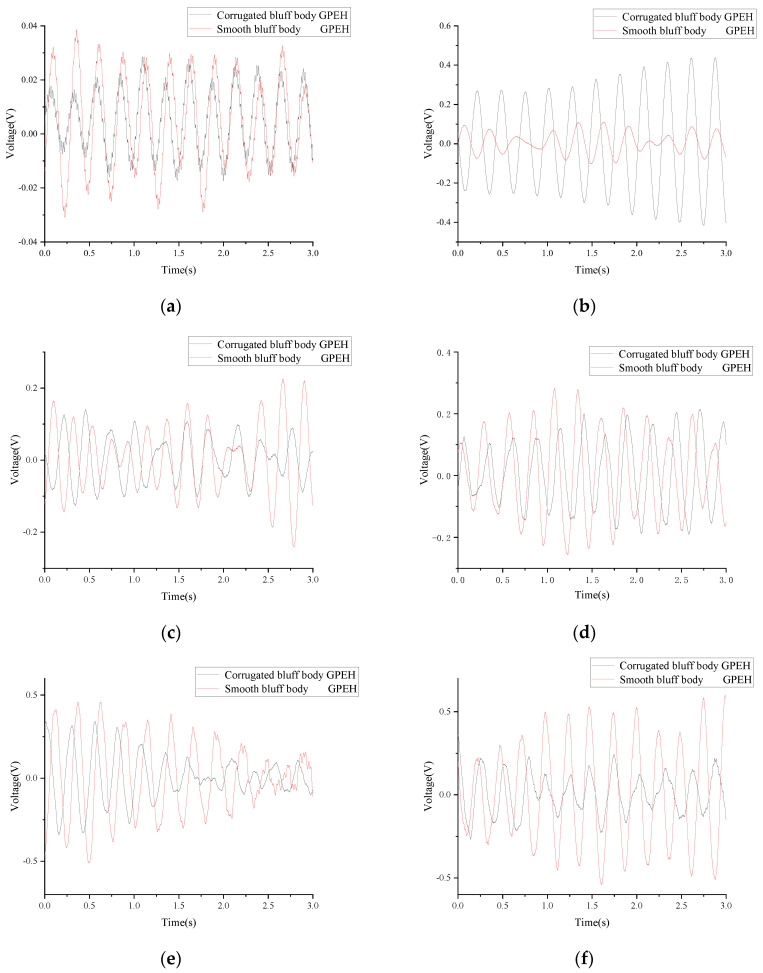
Output voltage of PZT film in the traditional GPEH and GPEH with corrugated bluff body. (**a**) The output voltage (*U* = 1 m/s). (**b**) The output voltage (*U* = 2 m/s). (**c**) The output voltage (*U* = 3 m/s). (**d**) The output voltage (*U* = 4 m/s). (**e**) The output voltage (*U* = 5 m/s). (**f**) The output voltage (*U* = 6 m/s).

**Figure 16 micromachines-14-01125-f016:**
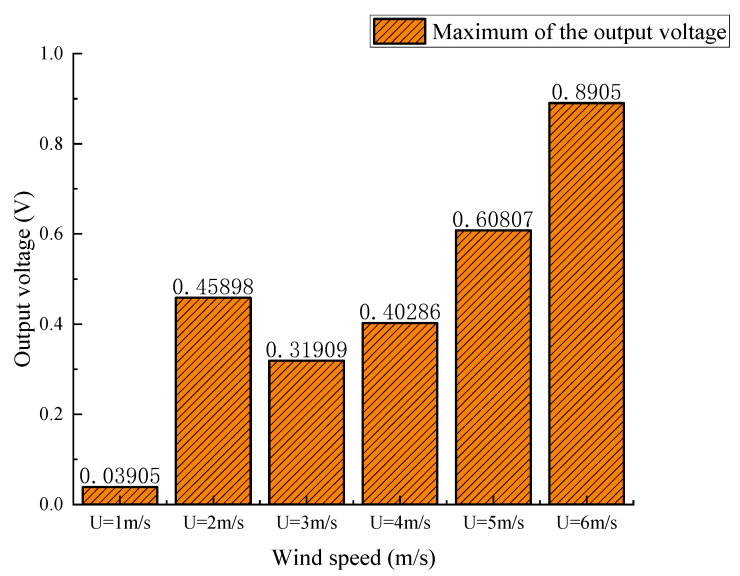
The maximum output voltage of the energy harvester based on corrugated bluff body under wind speed (*U* = 1~6 m/s).

**Figure 17 micromachines-14-01125-f017:**
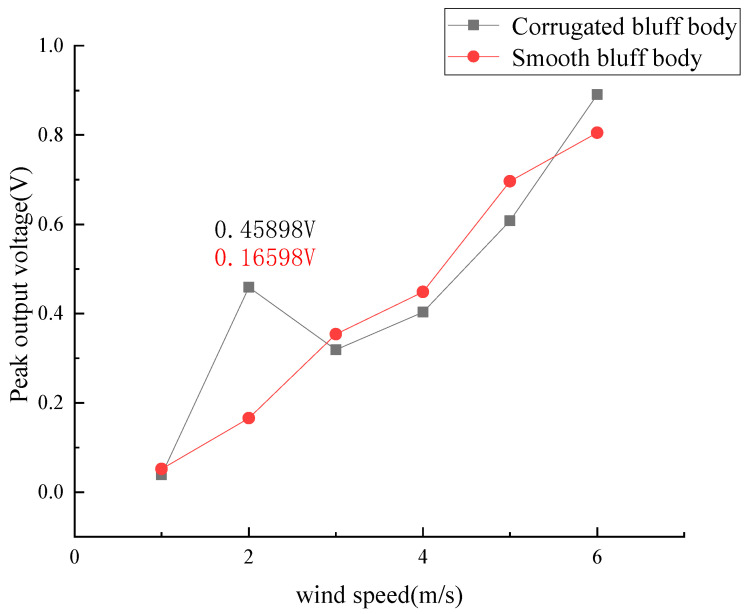
The peak output voltage with corrugated bluff body and smooth bluff body GPEHs.

**Figure 18 micromachines-14-01125-f018:**
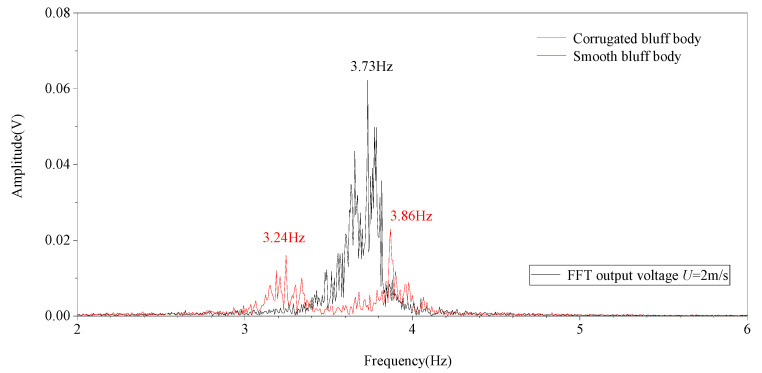
The frequency diagram of GPEHs with corrugated bluff body and smooth bluff body.

**Table 1 micromachines-14-01125-t001:** Parameters of numerical models.

Parameters	Values	Units
*U*	1	m/s
*W* _1_	800	mm
*L* _1_	4300	mm
*W* _2_	180	mm
*L* _2_	160	mm
*μ*	17.9 × 10^−6^	Pa·s
*ρ*	1.29	kg/m^3^
*V_inlet_*	6 × *U* × *y* × (*W* − *y*)/*W*^2^ × *step*1 (*t* [1/*s*])	m/s
*P_outlet_*	0	Pa

**Table 2 micromachines-14-01125-t002:** Initial conditions of CFD simulation.

Parameters	Values	Units
*T*	293.15	K
*P*	101,325	Pa
*C*	1400	J/(kg·K)
*λ*	0.023	W/(m·K)

*T* is the temperature of the air, *P* is the pressure of the air, *C* is the specific heat capacity of the air, and *λ* is the thermal conductivity of the air.

**Table 3 micromachines-14-01125-t003:** Parameters of PZT-5H.

Properties	Value/Units	Interpretation of Properties
K_P_ K_31_ K_33_ Kt	0.68 0.38 0.76 0.52	Coupling factors
ε^T^_r3_	3200	Dielectric constants
d_31_ d_33_ g_31_ g_33_	−275 × 10^−12^ C/N 620 × 10^−12^ C/N 9.7 × 10^−3^ vm/n 22 × 10^−3^ vm/n	Piezoelectric constant

## Data Availability

Not applicable.

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
