# Peer review of "Exploring the Potential of Flow-Induced Vibration Energy Harvesting Using a Corrugated Hyperstructure Bluff Body"

_micromachines, 2023, doi:10.3390/mi14061125_

Round 1

Reviewer 1 Report

This paper provides a galloping energy harvester design and discuss the performance on the use of corrugated hyperstructure bluff body. The author verified the feasibility of the energy harvester through a combination of simulation and experiment. The experiment was realized through a professional wind tunnel, which has strong persuasive power. However, the author's design and innovation are not outstanding, and it is difficult to see the significance of this work.

1.     In line 63, please check the grammar: When the wind is flow….

2.     In line 65, 66, 67 the noun. Vibration is not appropriate in this sentence and please check the grammar.

3.     In line 67, it should be “due to”.

4.     In line 73, what means by “periodic raised structure”?

5.     In line 76, please check the grammar.

6.     In line 170, the author claims that the lift force generated in the corrugated bluff body is 18.3% larger than traditional one. However, the author only calculates the maximum value. Why not choose the average value? In Figure 7, it seems the corrugated bluff body sometimes is less than traditional body. In figure 8, I also have the same question.

7.     What is the material of the corrugated bluff body used in this experiment as shown in Figure 11? Why choose this material and how do you design the size of the body? Will the weight of the body affect the results?

8.     In line 241, how do the author calculate the power of the energy harvester?

9.     Can you show the application of this energy harvester in a real scene?

10.  I think the reference list is too short for a research paper. I suggest the author including more related work.

There are too many grammatical errors in this paper. I suggest using some tools to improve the English quality, such as AI and translation apps. 

Author Response

Dear Reviewer,

Thank you for taking the time to review our work and provide your thoughtful feedback.  We greatly appreciate your effort in providing us with constructive criticism, and we assure you that we will take your comments into consideration to enhance the overall quality of our work.

We have attached a document containing the answer to your query.  We hope this helps clarify any confusion or doubts you may have had.  Should you require any further assistance or have any additional questions, please don't hesitate to reach out to us.

Once again, thank you for your valuable comments, and we look forward to continuing our cooperation in the future.

Best regards,
Yikai Yuan

Reviewer 2 Report

This paper explores the potential of flow-induced vibration energy harvesting using a corrugated hyperstructure bluff body. The following issues are suggested before the paper can be recommend for acceptance.

1. The reason for select the corrugated hyperstructure bluff body should be explained.

2. The CFD simulation should be better validated, e.g., in terms of the vibration amplitude and frequency.

3. More details of the CFD simulation should be provided, e.g., the turbulent model and the time integration method.

4. the Reynolds number of the wind tunnel test should be provided.

5. It is suggested to enrich the literature review with recent studies on flow-induce vibration-based energy harvesting, e.g., Effects of side ratio on energy harvesting from transverse galloping of a rectangular cylinder, Predefined angle of attack and corner shape effects on the effectiveness of square-shaped galloping energy harvesters, Piezoelectric energy harvesting from vortex-induced vibration of a circular cylinder: Effect of Reynolds number.

6. Careful proofreading is suggested.

Language should be improved.

Author Response

(The authors gave the same response as above.)

Round 2

Reviewer 1 Report

The authors have already answered my questions appropriately. However, I still think the English writing should be enhanced. Please check more times before final submission. 

Should be improved.

Reviewer 2 Report

The paper can be accepted in the present form.